# Multi-Techniques for Analyzing X-ray Images for Early Detection and Differentiation of Pneumonia and Tuberculosis Based on Hybrid Features

**DOI:** 10.3390/diagnostics13040814

**Published:** 2023-02-20

**Authors:** Ibrahim Abdulrab Ahmed, Ebrahim Mohammed Senan, Hamzeh Salameh Ahmad Shatnawi, Ziad Mohammad Alkhraisha, Mamoun Mohammad Ali Al-Azzam

**Affiliations:** 1Computer Department, Applied College, Najran University, Najran 66462, Saudi Arabia; 2Department of Artificial Intelligence, Faculty of Computer Science and Information Technology, Alrazi University, Sana’a, Yemen

**Keywords:** VGG16, ANN, ResNet18, tuberculosis, LBP, DWT, PCA, GLCM

## Abstract

An infectious disease called tuberculosis (TB) exhibits pneumonia-like symptoms and traits. One of the most important methods for identifying and diagnosing pneumonia and tuberculosis is X-ray imaging. However, early discrimination is difficult for radiologists and doctors because of the similarities between pneumonia and tuberculosis. As a result, patients do not receive the proper care, which in turn does not prevent the disease from spreading. The goal of this study is to extract hybrid features using a variety of techniques in order to achieve promising results in differentiating between pneumonia and tuberculosis. In this study, several approaches for early identification and distinguishing tuberculosis from pneumonia were suggested. The first proposed system for differentiating between pneumonia and tuberculosis uses hybrid techniques, VGG16 + support vector machine (SVM) and ResNet18 + SVM. The second proposed system for distinguishing between pneumonia and tuberculosis uses an artificial neural network (ANN) based on integrating features of VGG16 and ResNet18, before and after reducing the high dimensions using the principal component analysis (PCA) method. The third proposed system for distinguishing between pneumonia and tuberculosis uses ANN based on integrating features of VGG16 and ResNet18 separately with handcrafted features extracted by local binary pattern (LBP), discrete wavelet transform (DWT) and gray level co-occurrence matrix (GLCM) algorithms. All the proposed systems have achieved superior results in the early differentiation between pneumonia and tuberculosis. An ANN based on the features of VGG16 with LBP, DWT and GLCM (LDG) reached an accuracy of 99.6%, sensitivity of 99.17%, specificity of 99.42%, precision of 99.63%, and an AUC of 99.58%.

## 1. Introduction

All cells in our bodies need oxygen to live. The heart and lungs work in an integrated manner, called the pulmonary loop, to ensure that the body gets the proportion of oxygen in the blood that it needs [1]. The heart picks up blood that lacks enough oxygen and transports it to the lungs to purify it and provide it with a sufficient percentage of oxygen. The lungs also work to adapt the oxygen entering the body, make it commensurate with the body’s temperature and humidity level, and get rid of carbon dioxide by exhaling [2]. Lung diseases are classified into lung tissue diseases, bronchitis, and circulatory diseases that affect the lung. Lung tissue diseases and bronchitis cause limiting of the expansion of the lungs [3]. While diseases of the pulmonary circulation cause inflammation or thrombosis of the blood vessels, which causes the lungs to be unable to inhale and exhale normally [4]. Mycobacterium tuberculosis (TB) is a chronic and infectious disease of the lungs because of conjugated mycobacteria. Tuberculosis patients need a long-term treatment plan [5]. Tuberculosis is the leading cause of death among infectious diseases and the tenth cause of death among all diseases [6]. In 2020, more than ten million people contracted tuberculosis, and more than 1.5 million died worldwide [7]. There are many symptoms of tuberculosis, the most common of which are coughing, blood in the sputum, weight loss, fever, night sweats and chest pain. Additionally, there are symptoms such as chills, shortness of breath, fatigue and loss of appetite [8]. Early detection of tuberculosis is crucial for patients to take effectual treatment, increase the survival rate and prevent the further spread of Mycobacterium tuberculosis bacteria. Finding positive characteristics and biomarkers for tuberculosis patients in the early stage is difficult [9]. Therefore, early detection of tuberculosis requires a high level of integrated health care, as well as skilled physicians and radiologists. Therefore, in the first stage, if a proper diagnosis is made and appropriate treatment is received, patients can be treated clinically. Therefore, improving integrated health care is the basis for curing tuberculosis patients [10]. Pneumonia is a disease of the lungs caused by a bacterial and viral infection. Symptoms of pneumonia are similar to those of tuberculosis [11]. Doctors find that tuberculosis and pneumonia are diagnosed interchangeably because the clinical symptoms are similar [12]. Therefore, failing to distinguish between pneumonia and tuberculosis leads to patients receiving inappropriate treatment, a lower survival rate, a prolonged treatment period, and increased treatment costs [13]. When the clinical symptoms are similar, it is difficult to distinguish between the two diseases. Therefore, diagnostic imaging techniques have become the way to distinguish between the vital signs of the two diseases. X-ray imaging is the technique for diagnosing tuberculosis and pneumonia. Moreover, the X-ray images need radiologists and experts with experience and skill to detect the disease in the early stages and distinguish between tuberculosis and pneumonia. Therefore, artificial intelligence techniques using deep and machine learning are useful for accurate diagnosis in the early stages. Deep learning models provide helpful tools for extracting hidden features in X-rays that lead to the early diagnosis of pneumonia and tuberculosis. Experts and researchers have made efforts and time to diagnose pneumonia and tuberculosis at an early stage, in addition to differentiating between them with high accuracy. Therefore, the main objective of this study is to develop novel models for the differential diagnosis of pneumonia and tuberculosis. This study focused on extracting complex patterns from X-ray images using deep learning models and merging them; combining the features of deep learning with the features of texture, shape and geometry extracted using LDG algorithms.

The main contributions in this paper are as follows:
Enhancement of chest X-rays using overlapping of average and Laplacian filters;Reducing the high dimensionality deep features produced using the VGG16 and ResNet18;Application of a hybrid technology between deep learning models and the support vector machine (SVM) for early differentiation between pneumonia and tuberculosis;Fusing of the deep features of the VGG16 and ResNet18 before and after reducing the high dimensions to obtain more representative features of pneumonia and tuberculosis, which are then diagnosed by ANN;Early differentiation between pneumonia and tuberculosis by ANN through integrating the features of the VGG16 and ResNet18 separately with the hand-crafted features.

The rest of the work is ordered as follows: discussion of a set of prior studies in Section 2; applying materials for analyzing the X-ray images of a pneumonia and tuberculosis data set in Section 3; summarizing and displaying the results of the systems in Section 4; discussion on the performance of the proposed systems in Section 5; conclusions of the paper in Section 6.

## 2. Related Work

This section discusses several techniques from previous studies for diagnosing pneumonia and tuberculosis using artificial intelligence techniques.

Presented a system for distinguishing between pneumonia and tuberculosis using networks of stacked groups [14]. The system has two steps: the first is to choose the low-cost features for early diagnosis; the second is the selection of features with X-ray reports for differential diagnosis of pneumonia and tuberculosis. introduced an optimization algorithm to monitor the carrier wave ratio in analysis of the prevalence coefficient to assess the diagnosis of pests [15]. The method concluded that the images after noise reduction were apparent and the edges of the lung regular. introduced a hybrid model between VGG and a spatial transformer network and data augmentation with a CNN called VDSNet. The VDSNet achieved an accuracy of 73%; when using a sample of the data set instead of the entire data set, VDSNet requires less time and produces less accuracy [16]. developed DCNN by training the X-ray images of the tuberculosis data set and testing the system on another data set, which achieved an AUC of 70.54% [17]. In contrast, DCNN achieved an AUC of 98.45% with the same data set. Therefore, the DCNN system performs poorly when tested on another data set. [18] presented a new system that combines deep and hand-crafted features to classify tuberculosis [19]. Shilpa et al. presented the development of the U-Net++ method for lung segmentation and extracting affected areas for diagnosing tuberculosis and other lung diseases. The method achieved an accuracy of 95% for lung segmentation. presented the Siamese Deep to classify X-rays of pneumonia. The method divides each X-ray into two identical parts and compares the affected areas between the two parts to help doctors diagnose pneumonia [20]. presented a hybrid technique using AlexNet and SVM to detect pneumonia. The AlexNet model extracted deep features, then classified them using the SVM algorithm [21]. proposed reinforcement learning to diagnose and differentiate between pneumonia and tuberculosis. The fuzzy learning method was applied along with the wavelet to determine the acuteness of pneumonia and tuberculosis [22]. proposed CNNs, EfficientNet, and Vision Transformer to improve the performance of tuberculosis prediction from large data sets. Reinforcement learning techniques were also applied to achieve high diagnostic performance, reaching an accuracy of 97.72% [23]. proposed the SVM for diagnosing tuberculosis by extracting features from X-ray images. Features were extracted using six hand-crafted algorithms and nine deep learning models. The features were fed separately to the SVM classifier, which achieved the best accuracy of 92.5% [24]) presented the SVM algorithm for X-ray image diagnostics to detect tuberculosis through texture features [25]. The features were reduced, and the best features were selected using a genetic method and then fed into an SVM classifier for classification. Ref. [26] presented a TBXNet network for the X-ray classification of tuberculosis. The features are extracted from five double convolution layers; each has a different filter size ranging from size 32 to size 512. All convolution layers are merged with a pre-trained fusion layer. TBXNet achieved the best accuracy of 95.67%, precision of 95.1% and recall of 95.1%. Ref. [27] proposed a model for X-ray analysis of TB by combining the decisions of three learners using the combined ensemble method. The features were extracted from three pre-trained CNN models; then, the base learners were built by adding fully connected and softmax layers. Ref. [28] proposed an NLP technique for extracting TB labels from X-rays and comparing them to manual labels by experts. The data set was fed with both NLP and manual labels into the DCNN. The DCNN has achieved better results with manual labels than with NLP labels. Ref. [29] proposed a model for X-ray analysis of tuberculosis. Reaction–diffusion was applied to segment the lung in X-rays. Chan–Vese was used to isolate mediastinum from lung masks and extract the shape features of both mediastinum and lung masks. The MLP network achieved F-measures of 82.4% and 81% for discrimination between DS and MDR. Ref. [30] proposed an MBFAL network to analyze X-rays to detect pneumonia. The MBFAL network worked to identify pneumonia by integrating feature maps extracted from different branches to enhance the ability of the network to recognize the severity of pneumonia. The MBFAL network achieved an average accuracy of 95.61%. Ref. [31] proposed a deep learning model for the classification of X-rays for a multiclass data set for diagnosing pneumonia, tuberculosis and opacities. The architecture consists of a pre-trained VGG19 model followed by three blocks of convolutional and fully connected layers of CNN. The VGG19 + CNN construct achieved an accuracy of 96.48%, an F1 score of 95.62%, and a recall of 93.75%. Ref. [32] proposed a deep learning model for X-ray analysis of a multiclass data set for diagnosing pneumonia and tuberculosis. The X-rays are analyzed and explained using Grad-CAM activation maps, interpretable model interpretation (IMI), and shapely data from SHAP. Features are extracted for complex information. Predictive information from IMI, shapely data from SHAP, and Grad-CAM maps are used to explore hidden features using the DL model, which achieved an accuracy of 94.31%. Ref. [33] proposed a CNN with 22-layer and three machine learning methods to analyze X-rays for detecting pneumonia. The overfitting problem was addressed with a dense layer regulator. The features were extracted using CNN and classified using SVM and KNN, which achieved an accuracy of 97.32% and 96.55%, respectively.

From the above, we note that there is a lack of discriminatory studies for diagnosing pneumonia and tuberculosis. Because of the similarity of the symptoms between pneumonia and tuberculosis and the similarity of features in X-ray images, this study aims to extract the features from many methods and combine them with some, including hidden, features. In this study, features were extracted using VGG16 and ResNet18 and combined with features extracted using LDG algorithms; in addition to integrating the deep features of the VGG16 and ResNet18 together.

## 3. Materials and Methods

This section describes the materials and methods presented in this study to analyze the X-rays of pneumonia and tuberculosis data sets. The X-ray images were enhanced to obtain improved images and then fed into the VGG16 and ResNet18. VGG16 and ResNet18 produce high dimensional features, so the PCA reduces the high dimensional features. The first approach is a hybrid method for distinguishing between pneumonia and tuberculosis using VGG16 + SVM and ResNet-18 + SVM. The second approach distinguishes between pneumonia and tuberculosis using an ANN network, based on fusing the features of VGG16 and ResNet18 before and after dimensionality reduction. The third approach distinguishes between pneumonia and tuberculosis using ANN based on integrating deep features of VGG16 and ResNet18 separately with the hand-crafted features extracted by LDG algorithms as shown in Figure 1.

### 3.1. Description of the Data Set

This section describes the X-rays of pneumonia and tuberculosis data set collected to evaluate the systems. The data set contains 6892 X-rays collected from three sources from the Kaggle website, divided into three classes: the normal class, which contains 1583 normal X-rays, and 4273 X-rays for pneumonia patients, collected from Guangzhou Medical Center [34]. The third class represents tuberculosis, which contains 1036 X-ray images of tuberculosis patients collected from two sources: 336 X-rays in the Shenzhen data set from the US National Library of Medicine in Shenzhen Hospital [35]; and 700 X-rays collected from researchers at Qatar University in Doha and a group of doctors from Hamad Medical Corporation and the University of Dhaka, Bangladesh [36]. All X-rays are in PNG format, with a size of 512 × 512 pixels per image. Figure 2. shows a set of X-rays representing the three classes.

### 3.2. Enhancement of X-rays

X-ray images of patients with pneumonia and tuberculosis contain artifacts and noise caused by the patient’s movements when exposed to X-rays and offer various contrasts. In addition, collecting X-rays from multiple sources means that the images will feature the different levels of accuracy of the imaging devices. This causes different image homogeneity and poor CNN performance. Therefore, all X-rays have been enhanced to increase the contrast of the injury areas and remove noise before being fed into the CNN models. Two filters have been applied, the average filter to enhance the images and remove noise, and the Laplacian filter to highlight the contrast of the edges of the affected region [37].

First, the average filter is set to 5 * 5 pixels. The filter selects a central pixel, then calculates an average value for 24 adjacent pixels, and then replaces the average values with the center pixel. The filter continues until all the pixels in the X-ray are changed by the average of the neighboring pixels of each central pixel as shown in Equation (1). Thus, an enhanced X-ray is obtained.
(1)$$A\left(x\right)=\frac{1}{L}\sum_{i=0}^{L-1}y\left(x-i\right)$$
where $A\left(x\right)$ means the input, L means the image pixels and $y\left(x-i\right)$ means the previous input.

Second, the Laplacian filter is used, which increases the contrast of the affected areas and thus, improves the appearance of the edges of the region of interest (ROI), as in Equation (2).
(2)$$\nabla^{2}\;f=\frac{\partial^{2}\;f}{\partial^{2}\;x}+\frac{\partial^{2}\;f}{\partial^{2}\;y}$$
where $\nabla^{2}f$ means the second-order differential equation; *x*, *y* is the position of pixels in the matrix.

Finally, to obtain enhanced images with good edges, the images enhanced through the average filter are combined with the images enhanced through the Laplacian filter, as in Equation (3).
(3)$$\mathrm{Image}\;\mathrm{enhanced}\;=A\left(x\right)-\nabla^{2}f$$

It should be noted that the system selected random samples from the data set. Each image in Figure 2a is the same as Figure 2b after it has been enhanced.

### 3.3. Hybrid Models CNN with SVM

Pre-trained CNN models require high-spec computers and are time consuming to train a data set. Despite this, it does not achieve promising results for the differential diagnosis between pneumonia and tuberculosis. Therefore, the hybrid technologies VGG16 + SVM and ResNet-18 + SVM solve these challenges [38]. Hybrid techniques work on medium-cost hardware and are quick to train the data set and achieve better accuracy. The techniques operate in two stages: the first stage is a model VGG16 and ResNet18 to extract the features in high dimensions. The PCA reduces high dimensionality and stores low dimensional features in feature vectors. The second stage is to classify the low dimensional features using the SVM.

#### 3.3.1. Deep Feature Extraction

In recent years, CNN models have proven their superior capabilities in many fields, including health care, such as predicting outbreaks and disasters and the early diagnosis of diseases and tumors. CNN models consist of many diverse layers that have high capabilities to extract deep features without human intervention. After enhancing the X-ray images, they are passed to CNN models that process the images through complex calculations in many layers, each containing neurons associated with millions of weights and connections. Deep features are extracted by convolutional layers, pooling layers to reduce high dimensionality, and there are auxiliary layers that perform certain tasks.

Convolutional layers are the most critical layer in convolutional neural networks, and their name derives from the name of CNN networks. Each convolutional layer is given a particular task, for example, a layer that extracts geometric features, a layer that appears at the edges, a layer that extracts shape features, another layer that extracts color features, and so on. The three parameters that control the performance of the convolutional layer are filter size, p-step filter movement and zero padding [39]. A set of pixels is defined in the image according to the filter size *f* (*t*). The filter wraps around the image *x* (*t*) [40]. Each time a set of pixels is processed according to Equation (4), the filter wraps on the X-ray based on a p-step (*S*) parameter, and the size of the original X-ray is preserved by zero padding at the edges of the image [41].
(4)$$W\left(t\right)=\left(x*f\right)\left(t\right)=\int x\left(a\right)f\left(t-a\right)da$$
where *W*(*t*) means the output, *f*(*t*) means the filter and *x*(*t*) means the input.

Each convolutional layer produces a specific size of the processed image represented by the pixels, which is considered as an input to the next layer based on the input image size of the L × W × D layer, zero padding (*P*), filter size (*F*) and p-step (*S*), as described by Equation (5).
(5)$$O\left(N\right)=\frac{W-K+2P}{S}+1$$
where *O*(*N*) means the size of the output neuron.

Pooling layers: CNN models produce millions of parameters and connections that increase complex calculations, so CNN models provide pooling layers to solve this problem. Pooling layers reduce the high-dimensional features through two techniques: average and max-pooling layers [42]. The mechanism of the average pooling layer is as follows: the size of the average layer filter is set and based on the filter size, a set of image pixels is set, the average of the pixels has been computed, and then all the pixels are replaced by their average, as in Equation (6). The mechanism of max-pooling layers is as follows: based on the filter size, a set of image pixels is set, the max value is selected from the pixels, then all the pixels are replaced with a max value, as in Equation (7).
(6)$$z\left(i;\;j\right)=\frac{1}{k^{2}}\sum_{m,n=1\ldots .k}\;f\left[\left(i-1\right)p+m;\;\left(j-1\right)p+n\right]$$
(7)$$z\left(i;\;j\right)=max_{m,n=1\ldots .k}\;f\left[\left(i-1\right)p+m;\;\left(j-1\right)p+n\right]$$
where *m*, *n* denotes the position the feature matrix, *f* denotes the filter size, *p* means the step, and *k* means the size of the matrix.

Deep features are extracted from the X-ray images of the differentiation data set between pneumonia and tuberculosis by the VGG16 [43] and ResNet18 [44] models. The deep features that are produced with each model are high dimensional, so they are reduced by the PCA.

#### 3.3.2. SVM Classifier

The SVM solves classification tasks. The algorithm creates a line or hyperplane to categorize the features of the X-ray of pneumonia and tuberculosis with high efficiency. The margin is the distance between the closest data points between classes [45]. The hyperplane determines the performance of the algorithm; the higher the margin, the better the algorithm performs for separating the data set’s features into different classes [46]. The algorithm selects the best maximum margin that a hyperplane has with the maximum distance between class points. There are two types of algorithm work based on the number of data set classes: linear and non-linear. When a data set is separable, a linear SVM can solve these tasks. When the data set is not linearly separable, the SVM creates a kernel function. The kernel transforms the features of a data set from linearly inseparable to linearly separable. Figure 3 describes the structure of the hybrid method for diagnosing X-rays of pneumonia and tuberculosis.

When training an SVM based on the RBF kernel, there are two main parameters, C and gamma. Parameter C eliminates failures against the decision surface. The gamma parameter determines the effect of training with low and high values.

The VGG16 and ResNet18 [47] models receive the enhanced X-rays, then extract the features and save them as vectors. Due to the typical production by VGG16 and ResNet18 of high-dimensionality features, the PCA algorithm was implemented to reduce the high dimensionality and fed into the SVM for classification.

### 3.4. Integrating the Deep Features of the Two CNN Models

CNN models require expensive hardware, take a long time to train data, and do not achieve high accuracy, so the hybrid method using VGG16 and ResNet18 models with the ANN solves these challenges [48]. Also, the classification of features extracted by a single CNN model does not achieve promising levels of accuracy, so in this work, the features of the VGG16 and ResNet18 models were combined into the same feature vectors [49].

The steps in these strategies, as shown in Figure 4, consist of two proposed systems. The steps of the first proposed system: first, feeding the enhanced X-rays into the VGG16 model for deep feature extraction and storing them in a feature matrix with a size of 6892 × 2048. Second, feeding enhanced X-rays into the ResNet18 for feature extraction and storing them in a feature matrix where the size is 6892 × 2048. Third, the features of the VGG16 and ResNet18 are combined and stored in a feature matrix with a size of 6892 × 4096. Fourth, feeding the 6892 × 4096 feature matrix into the PCA to reduce high dimensionality [50], the PCA algorithm produces a low-dimensional feature matrix with a size of 6892 × 720. Fifth, feeding the 6892 × 720 feature matrix into the ANN algorithm to train the high-speed data set and classify it with promising levels of accuracy.

The ANN receives features from the PCA and processes them according to the required tasks. An ANN consists of an input layer, which receives the features and passes them on to the hidden layers. In this study, the input layer consists of input units with the same number of features inputted; the network consists of 15 hidden layers, which process the features using many complex arithmetic operations. Each hidden layer consists of many neurons connected by weights. The weights are adjusted based on the least error between the expected and actual input. The output layers consist of three neurons; each output neuron represents a class in the data set, where each image is labelled according to its appropriate class.

The steps of the second proposed system: the first and second steps are the same as the first proposed system. Third, feeding the feature matrix of the VGG16 model with a size of 6892 × 2048 and the feature matrix of the ResNet18 with a size of 6892 × 2048 to the PCA, separately. The PCA produces low-dimensional features and stores them in a two-feature matrix with a size of 6892 × 512 for each VGG16 and ResNet18 model, separately. Fourth, combining the features of both the VGG16 and ResNet18 into a new feature matrix with a size of 6892 × 1024. Fifth, feeding the feature matrix with a size of 6892 × 1024 to the ANN algorithm, which quickly trains the data set and classifies it with promising levels of accuracy.

The main approach idea is to check the performance of the ANN network based on merging the features of the VGG16 and the ResNet18 before and after applying the PCA.

### 3.5. Integrating CNN Features with Hand-Crafted Features

In this section, a new and novel technique is explored for X-ray diagnosis for the early diagnosis of pneumonia and tuberculosis and the differentiation between them by an ANN, based on fusing deep learning features with hand-crafted features extracted using LDG algorithms [51].

The steps of this technique: first, after enhancing the X-ray, they are fed into the VGG16 and ResNet18, separately. The VGG16 model extracts the features of the X-ray and stores them in a feature matrix of 6892 × 2048 in size. Similarly, the ResNet18 model extracts the features of the X-ray and stores them in a feature matrix of 6892 × 2048 in size. Second, feeding the feature matrix with a size of 6892 × 2048 for each model into the PCA to reduce the high-dimensional features, the PCA produces two low-dimensional feature matrices with a size of 6892 × 512 for each VGG16 and ResNet18 model. Third, the extraction of hand-crafted features using LDG algorithms; then combining the features of the LDG algorithms together in a feature matrix with a size of 6892 × 228.

Texture, geometric and shape features are among the most important features for representing the ROI selected in the X-ray. The LBP algorithm represents X-ray images in a gray contrast matrix. The LBP measures local contrast information in order to describe the texture of the X-ray surfaces [52]. In this work, the method was set to a size of 6 * 6, and according to this size, the algorithm selects a pixel in each iteration called the target (central) pixel ($g_{c}$) and the local contrast is analyzed on the basis of 35 adjacent pixels $(g_{p}$), as in Equation (8). The process is repeated according to the size of the matrix, and each central pixel is replaced by the value of the adjacent pixels, according to the LBP method. This method produces a feature matrix with a size of 6892 × 203.
(8)$${\mathrm{LBP}}_{R,P}=\sum_{p=0}^{P-1}s\left(g_{p}-g_{c}\right)2^{p}$$
where $g_{p}$ represents the adjacent pixels, $g_{c}$ represents the goal pixel, *P* means the neighboring pixel number, and *R* means the radius of the adjacent pixels.

The DWT method analyzes the ROI selected in the X-ray using low-/high-pass filters and divides the images into four sections for analysis. Low-pass filters analyze the first section of the X-ray and produce approximate parameters, from which three features are extracted through mean, variance and standard deviation measures [53]. While the low-high and high-low filters analyze two sections of the X-ray and produce detailed parameters, from each section, three features are extracted through mean, variance, and standard deviation measures. Finally, the high filter decomposes the last section of the X-ray image to produce detailed parameters, from which three features are extracted through the mean, variance and standard deviation measures. Thus, the algorithm produces 12 features, and they are stored in a feature matrix with a size of 6892 × 12.

The GLCM algorithm represents the ROI selected in the X-ray with a gray matrix. The algorithm checks each ROI in the X-ray to extract texture features based on spatial information. When the pixels of a specific region are almost the same, the texture of the region is smooth. In contrast, if the pixels of a specific region are various, the region’s texture is rough [54]. The algorithm examines the spatial information based on the distance d and four angles namely, 0°, 45°, 90° and 135° between the pixel and the neighbor. This algorithm produces 13 features based on statistical measures, which are stored in a feature matrix with a size of 6892 × 13.

Fourth, the low-dimensional deep features of the 6892 × 512 size extracted by the VGG16 model were combined with the hand-crafted features with the size of 6892 × 244, extracted by LDG algorithms together in feature vectors to make a size of 6892 × 740. Also, the low-dimensional deep features of the 6892 × 512 size extracted by the ResNet18 model were combined with the hand-crafted features with the size of 6892 × 244, extracted using the LDG algorithms together in feature vectors to make a size of 6892 × 740. Finally, the features of the VGG16 and LDG methods and the ResNet18 and LDG methods, whose size is 6892 × 740 for each method, were inputted into an ANN classifier to classify them into three classes namely, pneumonia, tuberculosis and normal, as in shown Figure 5.

## 4. Experimental Results of the Systems

### 4.1. Split of the Data Set

This study dealt with many hybrid techniques based on the fusion of features, which aims to extract the representative features of each X-ray to reach promising results for the early diagnosis of pneumonia and tuberculosis and to distinguish between them. The data set contains 6892 X-rays divided into three classes, 4273 X-rays for pneumonia patients, 1036 X-rays for tuberculosis patients, and 1583 X-rays for normal subjects. Moreover, 80% of the data set has been allocated to training and validating the systems and, the rest of the data set, 20% has been allocated for testing the performance of the systems. Table 1 describes the division of X-ray images for the data set of pneumonia and tuberculosis patients during all stages.

### 4.2. Evaluation Metrics

In this study, the performance of the systems for analyzing X-rays for early detection and distinguishing between them has been evaluated through measures of Equations (9)–(12); in addition to the AUC scale. Each measure is calculated based on correctly classified X-rays, called TP and TN, and incorrectly classified X-rays, called FP and FN. TP, TN, FP and FN are gained through the confusion matrix, which is created by each system as a quadrant matrix [55]. The AUC is calculated by the true positive rate, which is on the *y*-axis, against the false-positive rate, which is on the *x*-axis, as in Equation (13).
(9)$$\mathrm{Accuracy}=\frac{\mathrm{TN}+\mathrm{TP}}{\mathrm{TN}+\mathrm{TP}+\mathrm{FN}+\mathrm{FP}}\;*\;100\%$$
(10)$$\mathrm{Sensitivity}=\frac{\mathrm{TP}}{\mathrm{TP}+\mathrm{FN}}\;*\;100\%$$
(11)$$\mathrm{Specificity}=\frac{\mathrm{TN}}{\mathrm{TN}+\mathrm{FP}}\;*\;100\%$$
(12)$$\mathrm{Precision}=\frac{\mathrm{TP}}{\mathrm{TP}+\mathrm{FP}}\;*\;100\%$$
(13)$$\mathrm{AUC}\;=\frac{\mathrm{True}\;\mathrm{Positive}\;\mathrm{Rate}}{\mathrm{False}\;\mathrm{Positive}\;\mathrm{Rate}}$$
where TP refers to the X-rays of patients with pneumonia and tuberculosis correctly classified; TN refers to properly classified X-ray images of normal people; FP refers to X-rays of normal people, but who have been incorrectly classified as pneumonia and tuberculosis patients; FN refers to the X-rays of patients with pneumonia and tuberculosis, but who have been incorrectly classified as normal.

### 4.3. Data Set Balancing and Data Augmentation

CNN models need a big data set to train the models, which is a limitation consequent of the insufficiency of medical images. Also, an unbalanced data set is a limitation because the overall accuracy is inclined to the majority class in the data set. Therefore, CNN models provide a data augmentation technique that addresses the abovementioned challenges. It is shown in Figure 6 that before augmentation the data set is unbalanced and contains few X-rays, which are insufficient to train CNN models [56]. The data augmentation technique was used to obtain a balanced and sufficient data set to train the model. The data augmentation technique artificially increases the X-rays from the same classes of the data set by flipping, shifting and rotating at many angles. To balance the data set, the technique increases the X-rays in the minority classes by more than that in the majority classes. Table 2 describes the number of X-rays of pneumonia and tuberculosis in the data set before and after augmentation. It is noted that for the pneumonia class, each X-ray was increased by only one; for the tuberculosis class, the X-rays were increased by seven X-rays from each original X-ray; finally, the X-rays were increased by four X-rays from each original X-ray. Thus, the X-rays were increased and a balanced data set was obtained after augmentation, as in Figure 6.

### 4.4. Result of Hybrid Models CNN and SVM

This section discusses the performance of the hybrid systems, VGG16 + SVM and ResNet18 + SVM, for analyzing X-ray images for the early diagnosis of pneumonia and tuberculosis and distinguishing between them. The VGG16 and ResNet18 models receive X-ray data sets and analyze the images, extracting high-dimensional features and storing them in feature vectors. The PCA algorithm reduces high dimensionality and sends low-dimensional deep features into the SVM algorithm. The SVM algorithm quickly trains the data set and classifies it into three classes: pneumonia, tuberculosis, and normal.

Table 3 and Figure 7 summarize the results of VGG16 + SVM and ResNet18 + SVM for the diagnostic X-rays of the pneumonia and tuberculosis data set. It is noted that VGG16 + SVM is slightly outperformed by ResNet18 + SVM. The VGG16 + SVM technique achieved an accuracy of 97.5%, sensitivity of 97.24%, specificity of 98.76%, precision of 96.17% and an AUC of 97.16%. While the ResNet18 + SVM technique yielded an accuracy of 96.7%, sensitivity of 94.58%, specificity of 98.57%, precision of 95.2% and an AUC of 96.57%.

Figure 8 shows the confusion matrix generated using VGG16 + SVM and ResNet18 + SVM techniques to evaluate the X-ray data set and classify the images into three classes, normal, pneumonia and tuberculosis. The VGG16 + SVM achieved a diagnostic accuracy for each class level of 98.1%, 98.6% and 91.8% for the classification of normal, pneumonia and tuberculosis cases, respectively. While ResNet18 + SVM reached a diagnostic accuracy for each class level of 98.1%, 98.1% and 88.4% for the classification of normal, pneumonia and tuberculosis cases, respectively.

It is noted that there are cases of failures that the VGG16 + SVM technique could not properly classify, as follows: one X-ray in the normal class was classified as a pneumonia case; and five X-rays in the normal class were classified as tuberculosis cases. Four X-rays of pneumonia class were classified as normal cases; eight X-rays of pneumonia class were classified as tuberculosis cases. Nine X-rays in the tuberculosis class were classified as normal cases; eight X-rays in the tuberculosis class were classified as pneumonia cases. While the failures of the ResNet18 + SVM technique are as follows: a single X-ray in the normal class was classified as a pneumonia case; five X-rays in the normal class were classified as tuberculosis cases. Four X-rays in the pneumonia class were classified as normal cases; 12 X-rays in the pneumonia class were classified as tuberculosis cases. Nine X-rays in the tuberculosis class were classified as normal cases; 15 X-rays in the tuberculosis class were classified as pneumonia cases.

### 4.5. Result of Integrating the Deep Features of the Two CNN

This section discusses the performance of the ANN based on fusing the features of the VGG16 and ResNet18 before and after reducing the high-dimensional features of the X-ray analysis for the early diagnosis of pneumonia and tuberculosis and differentiating between them. The VGG16 and ResNet18 models receive the X-rays of pneumonia and tuberculosis data set, analyze the images, extract the high-dimensional features and save them in feature vectors. In these approaches, two methods were applied to combine the features of the VGG16 and ResNet18. First, the features of the VGG16 and ResNet18 were combined, and then the high-dimensional features were reduced by the PCA. Second, the high-dimensional features of the VGG16 and ResNet18 were reduced separately by the PCA and then combined. The ANN algorithm was fed the feature vectors from the two methods to quickly train it and accurately categorize the images into three categories: pneumonia, tuberculosis and normal.

Table 4 and Figure 9 summarize the performance results of the ANN based on deep feature merging of the VGG16 and ResNet18 models before and after deep feature dimension reduction. In the table, it is noted that ANN achieved better results when combining the deep features of the VGG16 and ResNet18 before reducing the high-dimensional features. The ANN based on feature fusion of the VGG16 and ResNet18 before applying the PCA algorithm achieved an accuracy of 98.5%, a sensitivity of 97.81%, a specificity of 99.24%, precision of 97.63% and an AUC of 98.36%. In contrast, the same network (ANN) based on the feature merger of the VGG16 and ResNet18 after applying the PCA algorithm yielded an accuracy of 97.8%, a sensitivity of 96.22%, a specificity of 98.4%, precision of 97.03% and an AUC of 98.26%. These approaches proved that merging the features of the VGG16 and ResNet18 models and then reducing the high dimensions is better than reducing the dimensions and then merging the features.

Figure 10 shows the confusion matrix using the ANN based on feature fusion of the VGG16 + ResNet18 before and after the PCA to evaluate the X-ray dataset and classify the images into three categories, normal, pneumonia and tuberculosis. It is worth noting that Class 0 represents normal cases, Class 1 represents pneumonia, and Class 2 represents tuberculosis. Based on deep feature fusion of the VGG16 + ResNet18 before reducing high-dimensional features, the ANN achieved diagnostic accuracy for each class of 98.7%, 99.3% and 95.2% for the classification of normal cases, tuberculosis and pneumonia, respectively. In contrast, the ANN based on the deep feature fusion of the VGG16 + ResNet18 after reducing the high-dimensional features reached a diagnostic accuracy for each class of 96.5%, 99.4% and 92.8% for the classification of normal cases, pneumonia and tuberculosis, respectively.

It is noted that there are cases of failures that the ANN based on deep feature fusion of the VGG16 + ResNet18 before reducing the high-dimensional features could not properly classify, as follows: four X-rays in the normal class were classified as tuberculosis cases. Two X-rays in the pneumonia class were classified as normal cases; four X-rays in the pneumonia class were classified as tuberculosis cases. Seven X-rays in the tuberculosis class were classified as normal cases; three X-rays in the tuberculosis class were classified as pneumonia cases. While the failures of the ANN based on deep feature fusion of the VGG16 + ResNet18 after reducing high-dimensional features are as follows: five X-rays in the normal class were classified as pneumonia cases; six X-rays in the normal class were classified as tuberculosis cases. Five X-rays in the pneumonia class were classified as tuberculosis cases. Six X-rays in the tuberculosis class were classified as normal cases; nine X-rays in the tuberculosis class were classified as pneumonia cases.

### 4.6. Result of Integrating CNN Features with Hand-Crafted Features

This section discusses the ANN results based on merging the features of the VGG16 model with hand-crafted features, and merging the features of the ResNet18 model with hand-crafted features, for analyzing X-ray images for the early diagnosis of pneumonia and tuberculosis and differentiating between them. The VGG16 and ResNet18 models receive the X-rays from the pneumonia and tuberculosis data set for analysis and deep feature extraction. In these techniques, two methods have been applied to combine the deep and hand-crafted features. First, the features of the VGG16 model have been combined with the features of the LDG algorithms. Second, the features of the ResNet18 model have been combined with the features of the LDG algorithms. The ANN algorithm has been fed the feature vectors for the two methods to train and accurately classify the images into three classes: pneumonia, tuberculosis and normal. Some of the proposed systems’ evaluation tools for the pneumonia and tuberculosis data set will be discussed.

#### 4.6.1. Error Histogram

When evaluating the pneumonia and tuberculosis data set, the error histogram is a tool for evaluating the ANN for diagnosing the X-rays of the pneumonia and tuberculosis data set. The network records the error rates during the training, validation and testing phases. Each stage is marked with a specific color, where blue represents the training of the data set, green represents validation and weight adjustment, red represents the testing of new samples, and orange represents the zero value between the target and output values, which is the best achievement of a network. Figure 11 shows the network evaluation for evaluating a data set for pneumonia and tuberculosis. The ANN, with features of the VGG16 and LDG, achieved the best results between values −0.9401 and 0.9401. In contrast, the ANN, with features of the ResNet18 and LDG, achieved the best results between values −0.9491 and 0.95.

#### 4.6.2. Best Validation Performance

When evaluating the pneumonia and tuberculosis data set, cross-entropy is a tool for evaluating the ANN for diagnosing X-rays. This tool measures the rate of difference between actual and outputs during each period. The network records the minimum error during all phases. Each stage is marked with a specific color, blue represents the training of the data set, green represents validation and adjusting weights, and red represents the testing of new samples. Figure 12 shows the network assessment of the data set for pneumonia and tuberculosis. Based on the features of the VGG16 and LDG, the ANN achieved the best results at epoch 19 with a minimum error of 0.00078734. On the other hand, the ANN, with features of the ResNet18 and LDG, achieved the best results at epoch 39, with a minimum error of 0.0091785.

#### 4.6.3. Gradient and Validation Checks (GVC)

The GVC is an ANN performance assessment tool for analyzing the pneumonia and tuberculosis data set. The network measures the GVC in each epoch during the implementation of the network. In each epoch, the network records the GVC by saving the minimum error between the actual and expected values. Figure 13 shows the performance of the ANN with X-ray images from the pneumonia and tuberculosis data set. The ANN based on the VGG16 and LDG features reached the best results at the gradient of 0.00056044 in the 45 epoch and with a validation of 6. On the other hand, the ANN based on the ResNet18 and LDG features reached the best results at the gradient of 0.0021944 in epoch 45 and with a validation of 6.

#### 4.6.4. Confusion Matrix

The confusion matrix is one of the most critical evaluation criteria for the performance of artificial intelligence systems. In this study, the ANN algorithm evaluated the X-ray images of the pneumonia and tuberculosis data set based on combining the CNN features with the hand-crafted features. Due to the similar features of pneumonia and tuberculosis, especially in the early stages, the features play a crucial role in distinguishing between the two diseases. This section focuses on extracting the deep features of the VGG16 and ResNet18 and fusing them with the features of shape, texture and geometry extracted using LDG algorithms. Thus, two feature matrixes with a size of 6892 × 740 were extracted for each of the VGG16 and LDG, and the ResNet18 and LDG. The two feature matrixes were inputted into the ANN to classify the X-rays of the pneumonia and tuberculosis data set into three classes: normal, pneumonia and tuberculosis.

Figure 14 shows the confusion matrix generated by the ANN algorithm with the features of the VGG16 and LDG, and the ResNet18 and LDG. The confusion matrix is a quaternary form matrix that represents correctly and incorrectly classified samples. Where the primary diameter of the confusion matrix represents X-rays correctly classified, and the rest of the confusion matrix represents X-rays incorrectly classified. When diagnosing hybrid features for the VGG16 and LDG, the ANN achieved an accuracy of 99.6% and an accuracy at the level of each class of 100%, 99.9% and 97.6% for the classification of normal cases, tuberculosis and pneumonia, respectively. In contrast, when diagnosing hybrid features for the ResNet18 and LDG, the ANN achieved an accuracy of 99.5% and an accuracy at the level of each class of 99.7%, 99.9% and 97.6% for the classification of normal cases, tuberculosis and pneumonia, respectively.

It is noted that there are cases of failures when using the ANN with the features of the VGG16 and LDG, which incorrectly classified the following: one X-ray in the pneumonia class was classified as a normal case. One X-ray in the tuberculosis class was classified as a normal case; four X-rays in the tuberculosis class were classified as pneumonia cases. While the failures of the ANN with the fusion features of the ResNet18 and LDG incorrectly classified the following: one X-ray in the normal class was classified as a pneumonia case. One X-ray in the pneumonia class was classified as a tuberculosis case. One X-ray in the tuberculosis class was classified as a normal case; four X-rays in the tuberculosis class were classified as pneumonia cases.

The technique of integrating the CNN features with hand-crafted features, to diagnose the X-ray images of pneumonia and tuberculosis and distinguish between them, is one of the best methods to represent the features of each image accurately. Table 5 and Figure 15 show the results of an ANN with the fusion of the CNN features with features extracted using LDG algorithms. The ANN with the features of the VGG16 and LDG reached an accuracy of 99.6%, a sensitivity of 99.17%, a specificity of 99.42%, precision of 99.63% and an AUC of 99.58%. In contrast, the same classifier (ANN) with the features of the ResNet18 and LDG reached an accuracy of 99.5%, a sensitivity of 99.16%, a specificity of 99.7%, precision of 99.53% and an AUC of 99.59%.

## 5. Discussion and Comparison of the Achievement of the Systems

In this study, several systems were developed that focused mainly on integrating the features extracted by many methods for diagnosing X-ray images of pneumonia and tuberculosis and distinguishing between them. All X-rays were subjected to enhancement procedures to increase the contrast and remove noise. This study is divided into three approaches, each with two proposed systems. The first approach is two systems, VGG16 + SVM and ResNet18 + SVM, which is a two-block hybrid method: the first block is the VGG16 and ResNet18 for feature extraction. The second block is the SVM algorithm that receives the features and classifies them with a high level of accuracy. The VGG16 + SVM and the ResNet18 + SVM achieved an accuracy of 97.5% and 96.7%, respectively.

The second proposed approach to the analysis of X-ray images to differentiate pneumonia and tuberculosis using an ANN based on integrating deep features of the VGG16 and ResNet18 before and after reducing the high-dimensional features. The ANN, when combining the deep features of the VGG16 and ResNet18 before dimensional reduction, obtained better results than merging the features after dimensional reduction. The ANN achieved an accuracy of 98.5% and 97.8% based on the feature combining of the VGG16 and ResNet18 models before and after the PCA algorithm, respectively. The third proposed approach to diagnostic X-rays to differentiate pneumonia and tuberculosis using an ANN based on integrating deep features of the VGG16 and ResNet18 models separately with the hand-crafted features extracted by LDG algorithms. The ANN achieved an accuracy of 99.6% based on the VGG16 and LDG features, and 99.5% based on the ResNet18 and LDG features. It is noted that the best results were obtained when combining the features of the CNN with the shape, texture and geometry features, which were extracted by LDG algorithms.

Table 6 summarizes the performance of all the proposed systems for diagnosing X-ray images of pneumonia and tuberculosis and distinguishing between them. All the systems note the diagnostic accuracy of each class in the pneumonia and tuberculosis data set. With the hybrid features of the VGG16 and LDG, the ANN achieved the best overall accuracy of 99.6%. For the pneumonia class, the best accuracy of 100% was achieved using the ANN with hybrid features of the VGG16 and LDG. For the tuberculosis class, the ANN with the hybrid features of the VGG16 and LDG, and the hybrid features of the ResNet18 and LDG, achieved an equal accuracy of 99.9%.

Figure 16 displays the execution achieved by all the proposed methods for diagnosing X-rays of the pneumonia and tuberculosis data set and distinguishing between them.

## 6. Conclusions

Tuberculosis is one of the most contagious and rapidly spreading diseases, so early diagnosis is crucial to reduce the spread and for patients to receive appropriate treatment. Because of the similar symptoms and characteristics of pneumonia and tuberculosis, accurate diagnosis is challenging for radiologists and physicians. Some techniques help diagnosis, the most prominent of which is the chest X-ray technique. However, because of the similar characteristics and vital signs between pneumonia and tuberculosis, it is challenging for doctors to diagnose early and distinguish between pneumonia and tuberculosis. Thus, artificial intelligence techniques solve this challenge by developing automated computer-aided techniques for early differentiation between pneumonia and tuberculosis. Because of the similarities between pneumonia and tuberculosis, this study focuses on the fusion feature of deep learning and LDG methods. This study consists of three approaches, each with two proposed systems for early differentiation between pneumonia and tuberculosis. The first proposed approach is to distinguish between pneumonia and tuberculosis using hybrid techniques, VGG16 + SVM and ResNet18 + SVM. The techniques consist of the VGG16 and ResNet18 models for feature extraction and an SVM classifier for deep feature classification. The second proposed approach for early differentiation between pneumonia and tuberculosis using ANN is based on integrating deep features of the VGG16 and ResNet18 before and after the PCA. The third proposed approach for early differentiation between pneumonia and tuberculosis using ANN is based on integrating features of the VGG16 and ResNet18 separately with hand-crafted features, producing two feature matrixes extracted using the VGG16 and LDG, and the ResNet18 and LDG. All systems reached promising results in early diagnosis and differentiation between pneumonia and tuberculosis. The ANN based on the features of the VGG16 and LDG reached an accuracy of 99.6%, a sensitivity of 99.17%, a specificity of 99.42%, precision of 99.63% and an AUC of 99.58%.

## Figures and Tables

**Figure 1 diagnostics-13-00814-f001:**
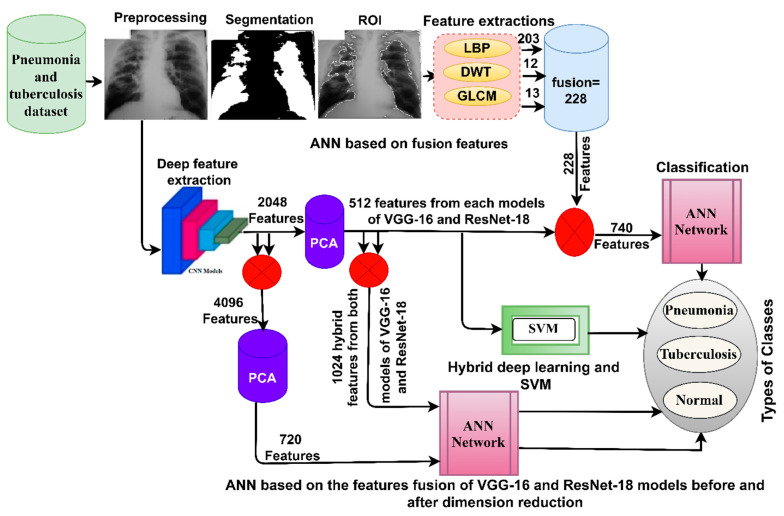
The framework of the structure for the proposed systems for the diagnosis of X-rays of pneumonia and tuberculosis, and for distinguishing between them.

**Figure 2 diagnostics-13-00814-f002:**
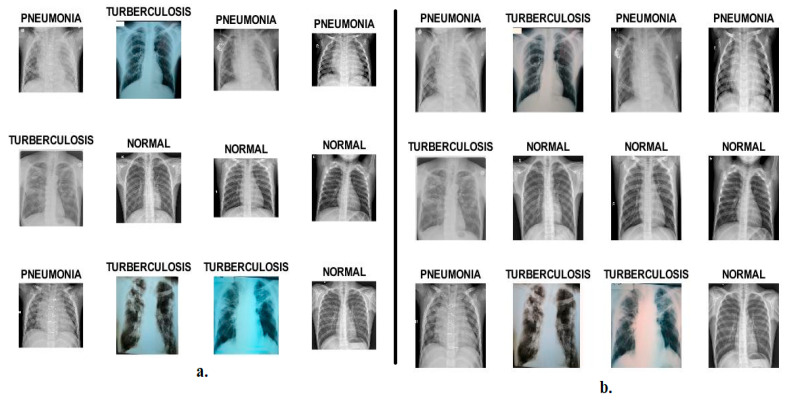
Samples from the pneumonia and tuberculosis data set. (**a**). Before improving the X-rays. (**b**). After improving the X-rays.

**Figure 3 diagnostics-13-00814-f003:**
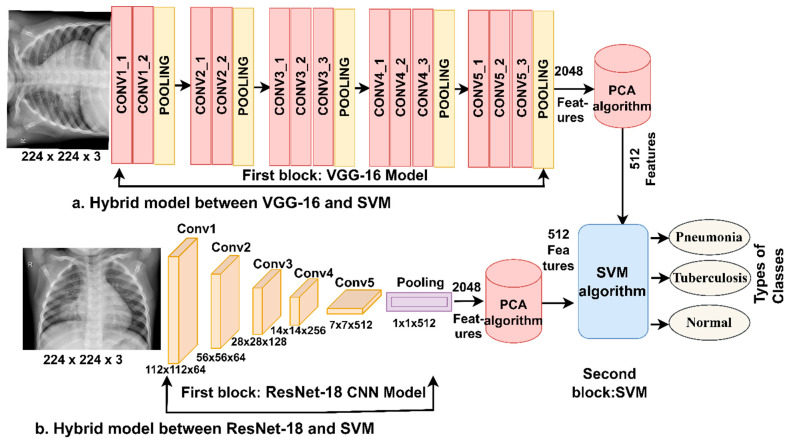
Structure framework of the hybrid system for early diagnosis and discrimination between pneumonia and tuberculosis.

**Figure 4 diagnostics-13-00814-f004:**
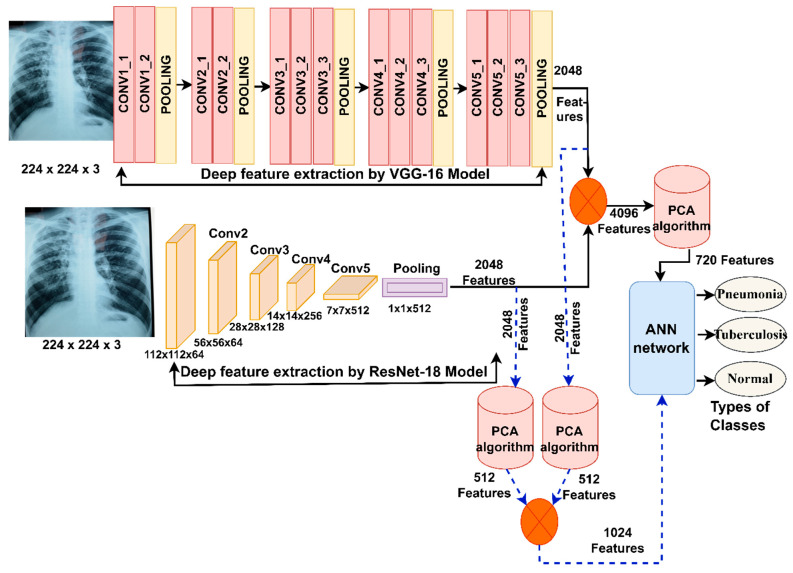
The structure of the early diagnosis technique and the discrimination of pneumonia and tuberculosis by ANN, based on the fusion of CNN features.

**Figure 5 diagnostics-13-00814-f005:**
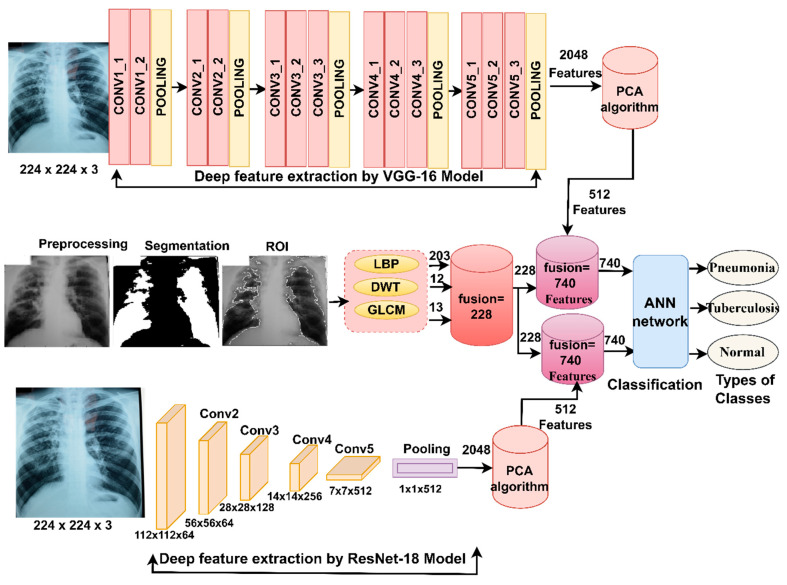
Structure framework of the technique for early diagnosis and discrimination of pneumonia and tuberculosis by ANN, based on the fusion of CNN features with hand-crafted features.

**Figure 6 diagnostics-13-00814-f006:**
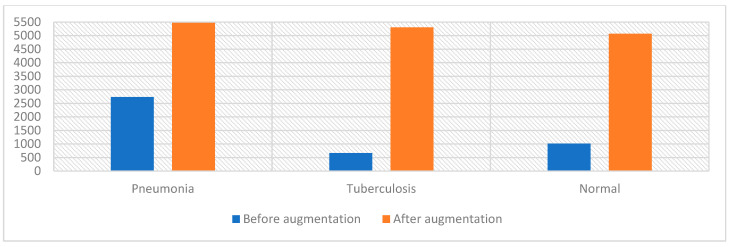
The number of X-rays of pneumonia and tuberculosis in the data set before and after applying the data augmentation technique.

**Figure 7 diagnostics-13-00814-f007:**
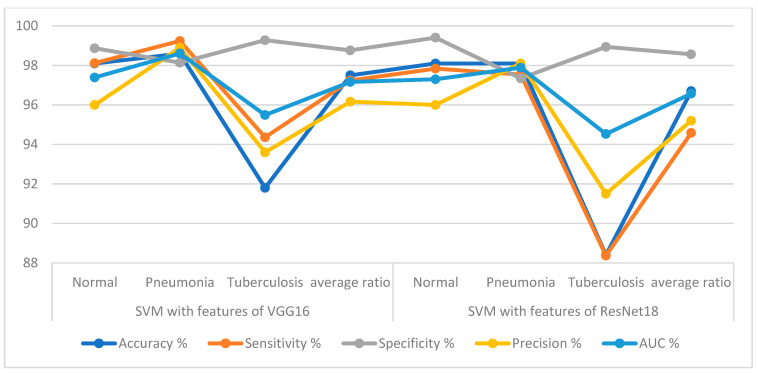
The performance of the hybrid methods for diagnosing the X-rays of the pneumonia and tuberculosis data set.

**Figure 8 diagnostics-13-00814-f008:**
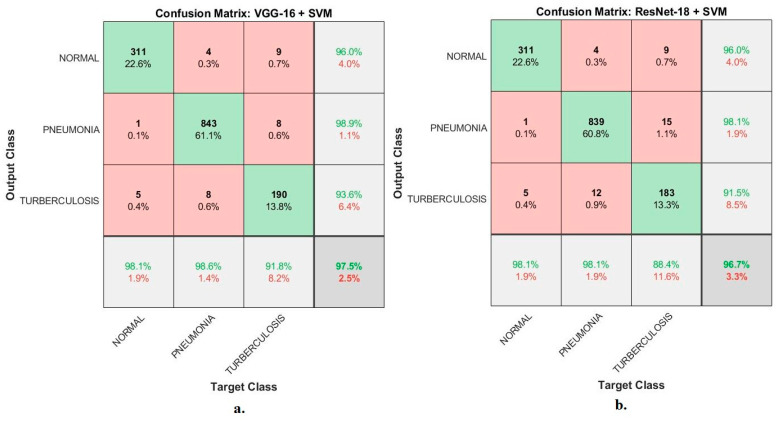
The confusion matrix displays the X-ray results for diagnosing pneumonia and tuberculosis using (**a**). VGG16 + SVM; (**b**). ResNet18 + SVM.

**Figure 9 diagnostics-13-00814-f009:**
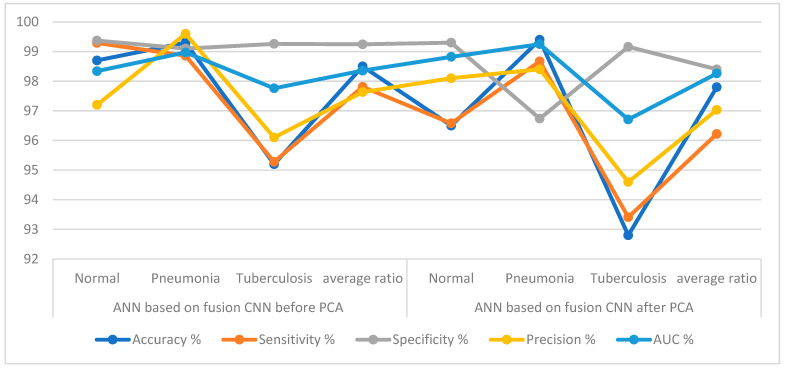
The ANN performance based on integrating the features for the X-ray diagnostics of the pneumonia and tuberculosis data set.

**Figure 10 diagnostics-13-00814-f010:**
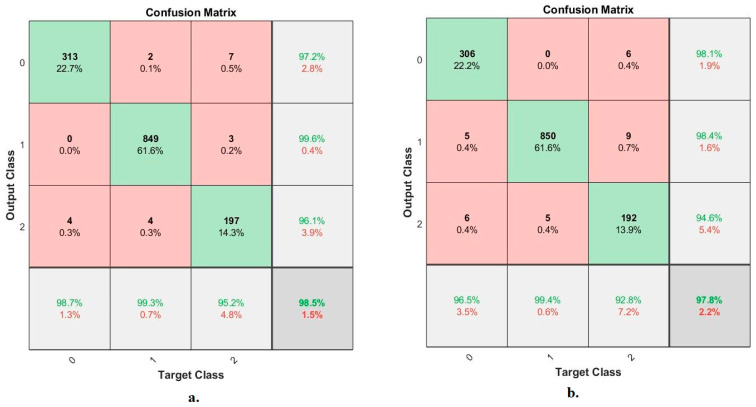
Confusion matrix displaying the ANN results for the diagnosis of X-rays of pneumonia and tuberculosis based on deep feature integration, (**a**). VGG16 + ResNet18 before PCA algorithm; (**b**). VGG16 + ResNet18 after PCA algorithm.

**Figure 11 diagnostics-13-00814-f011:**
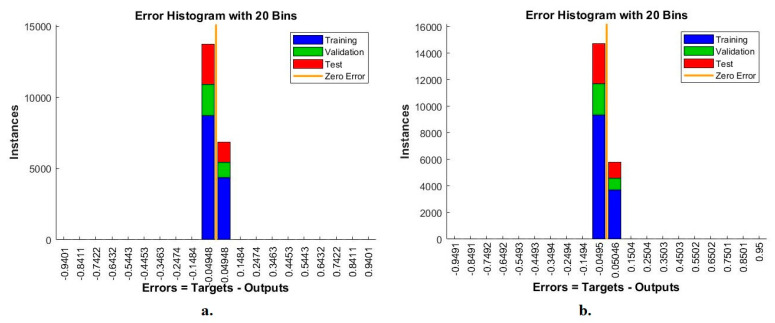
The error histogram for diagnosing the X-ray images of pneumonia and tuberculosis using the ANN with features (**a**). VGG16 and LDG; (**b**). ResNet18 and LDG.

**Figure 12 diagnostics-13-00814-f012:**
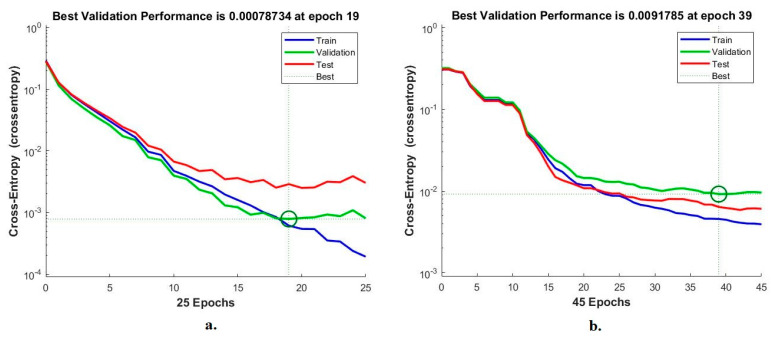
The best validation performance for diagnosing X-ray images of pneumonia and tuberculosis using the ANN with features (**a**). VGG16 and LDG; (**b**). ResNet18 and LDG.

**Figure 13 diagnostics-13-00814-f013:**
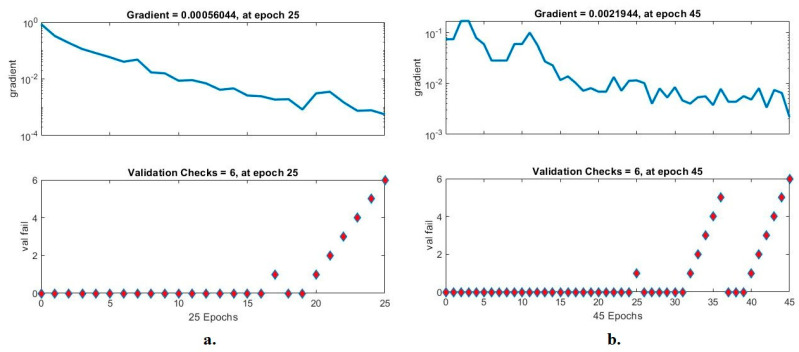
The GVC for diagnosing X-ray images of pneumonia and tuberculosis using the ANN with features (**a**). VGG16 and LDG; (**b**). ResNet18 and LDG.

**Figure 14 diagnostics-13-00814-f014:**
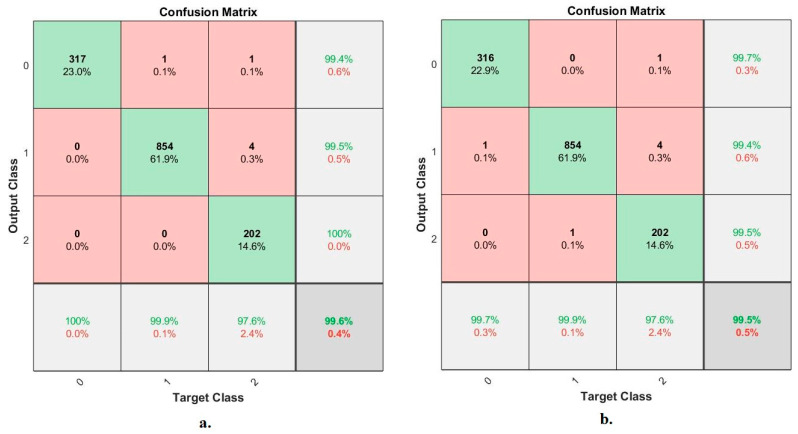
Confusion matrix for diagnosing the X-ray images of pneumonia and tuberculosis using the ANN with features (**a**). VGG16 and LDG; (**b**). ResNet18 and LDG.

**Figure 15 diagnostics-13-00814-f015:**
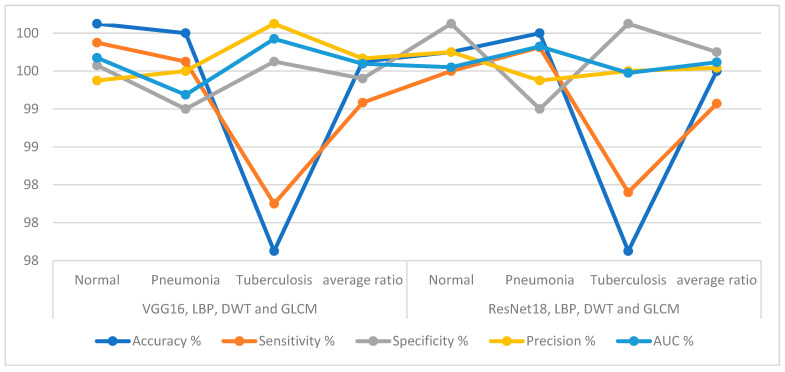
ANN achievement based on fusing the features of CNN with hand-crafted features for diagnosing the X-rays of the pneumonia and tuberculosis data set.

**Figure 16 diagnostics-13-00814-f016:**
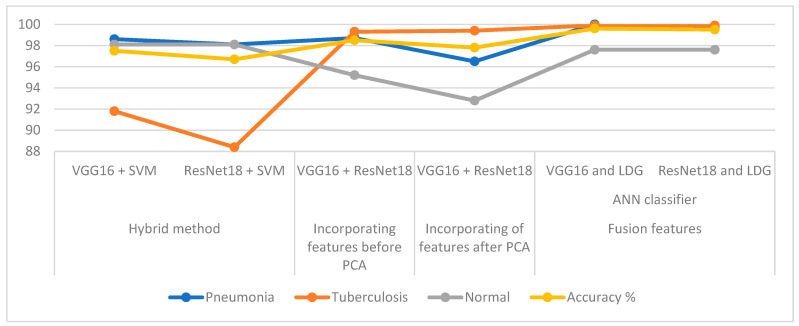
The execution and differentiation of the proposed methods for diagnosing the X-rays of the pneumonia and tuberculosis data set.

**Table 1 diagnostics-13-00814-t001:** Splitting of pneumonia and tuberculosis data set during implementation phases.

Phase	80:20		Testing Phases 20%
Classes	Training (80%)	Validation (20%)	
Pneumonia	2734	684	855
Tuberculosis	663	166	207
Normal	1013	253	317

**Table 2 diagnostics-13-00814-t002:** Method of data augmentation of the X-rays for pneumonia and tuberculosis data set to balance the data set.

Phase	Training Phase		
Classes	Pneumonia	Tuberculosis	Normal
Before augmentation	2734	663	1013
After augmentation	**5468**	**5304**	**5065**

**Table 3 diagnostics-13-00814-t003:** Results of the CNN with the SVM technique for diagnosing the pneumonia and tuberculosis data set and distinguishing between them.

Techniques	Type of Class	Accuracy %	Sensitivity %	Specificity %	Precision %	AUC %
SVM with features of VGG16	Normal	98.1	98.12	98.87	96	97.39
	Pneumonia	98.6	99.24	98.14	98.9	98.61
	Tuberculosis	91.8	94.36	99.28	93.6	95.49
	**average ratio**	**97.50**	**97.24**	**98.76**	**96.17**	**97.16**
SVM with features of ResNet18	Normal	98.10	97.84	99.41	96.00	97.30
	Pneumonia	98.10	97.54	97.35	98.10	97.89
	Tuberculosis	88.40	88.37	98.94	91.50	94.53
	**average ratio**	**96.70**	**94.58**	**98.57**	**95.20**	**96.57**

**Table 4 diagnostics-13-00814-t004:** Results of the ANN network based on fused features between the VGG16 and ResNet18 models.

Techniques	Type of Class	Accuracy %	Sensitivity %	Specificity %	Precision %	AUC %
ANN based on fusion CNN before PCA	Normal	98.7	99.29	99.37	97.2	98.34
	Pneumonia	99.3	98.86	99.1	99.6	98.97
	Tuberculosis	95.2	95.28	99.26	96.1	97.76
	**average ratio**	**98.50**	**97.81**	**99.24**	**97.63**	**98.36**
ANN based on fusion CNN after PCA	Normal	96.50	96.58	99.30	98.10	98.82
	Pneumonia	99.40	98.67	96.74	98.40	99.25
	Tuberculosis	92.80	93.41	99.16	94.60	96.71
	**average ratio**	**97.80**	**96.22**	**98.40**	**97.03**	**98.26**

**Table 5 diagnostics-13-00814-t005:** ANN performance based on the fusion of CNN features with hand-crafted features.

Techniques	Type of Class	Accuracy %	Sensitivity %	Specificity %	Precision %	AUC %
VGG16, LBP, DWT and GLCM	Normal	100	99.80	99.56	99.40	99.64
	Pneumonia	99.90	99.60	99.10	99.50	99.25
	Tuberculosis	97.60	98.10	99.60	100	99.84
	**average ratio**	**99.60**	**99.17**	**99.42**	**99.63**	**99.58**
ResNet18, LBP, DWT and GLCM	Normal	99.70	99.50	100	99.70	99.54
	Pneumonia	99.90	99.75	99.10	99.40	99.76
	Tuberculosis	97.60	98.22	100	99.50	99.48
	**average ratio**	**99.50**	**99.16**	**99.70**	**99.53**	**99.59**

**Table 6 diagnostics-13-00814-t006:** Results of the proposed systems in this study for diagnosing pneumonia and tuberculosis and distinguishing between them.

Techniques		Classes	Pneumonia	Tuberculosis	Normal	Accuracy %
Hybrid method		VGG16 + SVM	98.6	91.8	98.1	97.5
		ResNet18 + SVM	98.1	88.4	98.1	96.7
Incorporating features before PCA		VGG16 + ResNet18	98.7	99.3	95.2	98.5
Incorporating features after PCA		VGG16 + ResNet18	96.5	99.4	92.8	97.8
Fusion features	ANN classifier	VGG16 and LDG	100	99.9	97.6	99.6
		ResNet18 and LDG	99.7	99.9	97.6	99.5

## Data Availability

The data supporting the proposed systems were collected from a publicly available data set at the following links: https://www.kaggle.com/datasets/paultimothymooney/chest-xray-pneumonia (accessed on 10 October 2022); https://www.kaggle.com/datasets/kmader/pulmonary-chest-xray-abnormalities?select=Montgomery (accessed on 10 October 2022); https://www.kaggle.com/datasets/tawsifurrahman/tuberculosis-tb-chest-xray-dataset (accessed on 10 October 2022).

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
