# Peer review of "Multi-Techniques for Analyzing X-ray Images for Early Detection and Differentiation of Pneumonia and Tuberculosis Based on Hybrid Features"

_diagnostics, 2023, doi:10.3390/diagnostics13040814_

Round 1
Reviewer 1 Report
Comments for "Multi-Techniques for Analyzing X-ray images for Early Detection and Differentiation of Pneumonia and Tuberculosis Based on Hybrid Features"
1. The figure quality should be improved, the low pixel of figures make the content unable to read,such as Fig10
2. The method should be available as open source (github) for scientific reproducibility.
3. there are too many figuires in the main text with similar purpose, eg figure 7 and 9, so authors should consider combin some figures or move them in supplementary materials
Author Response
Respected Professors
Responses are attached

Reviewer 2 Report
In this manuscript, the authors proposed three methods to differentiate patients with pneumonia and tuberculosis through chest X-ray imaging. In all three methods, the authors firstly used well known VGG16 and ResNet18 networks to extract features. In one of methods, the authors also applied handcrafted algorithms, including LBP, DWT, GLCM algorithms to extract features. Then SVM and neural network were used to classify patients based on features extracted by different methods. Among three methods, using neural network to classify features extracted from VGG/ResNet network and handcrafted algorithm shows the best performance. Overall, this manuscript is interesting. Some information is missing and needs further clarification. I recommend major revision on this manuscript. Below are comments:
1. Some acronyms were not clearly defined in the introduction.
2. The contrast and intensity of raw X-ray images from different sources seem different. After applying average filter and Laplacian filter, the image qualities appear better. However, it is better to show statistics from different image sources. The superior classification result may be related to the different categories from different image sources.
3. What kernel was applied in SVM and what about regularization parameters in SVM?
4. It is not clear what ANN network was used for classification.
5. Many figures are in low resolution. High resolution images are needed.
Author Response

(The authors gave the same response as above.)

Round 2
Reviewer 2 Report
The revision looks good to me. The manuscript can be accepted.